# Characteristics and management of adolescents attending the ED with fever: a prospective multicentre study

Dorine Borensztajn ,[1] Nienke N Hagedoorn ,[1] Enitan Carrol,[2] Ulrich von Both ,[3] Juan Emmanuel Dewez ,[4] Marieke Emonts,[5] Michiel van der Flier,[6] Ronald de Groot,[7] Jethro Herberg,[8] Benno Kohlmaier,[9] Michael Levin,[8] Emma Lim,[5] Ian Maconochie,[8] Federico Martinon Torres,[10] Ruud Nijman,[8] Marko Pokorn,[11] Irene Rivero-Calle,[10] Maria Tsolia,[12] Clementien Vermont,[13] Dace Zavadska,[14] Werner Zenz,[9] Joany Zachariasse,[1] Henriette A Moll ,[1] On behalf of PERFORM Consortium: Personalised Risk assessment in febrile children to optimise Real-life Management across the European Union

**Correspondence to**
Dorine Borensztajn;
d.borensztajn@erasmusmc.nl

## ABSTRACT

**Objective** Most studies on febrile children have focused on infants and young children with serious bacterial infection (SBI). Although population studies have described an increased risk of sepsis in adolescents, little is known about febrile adolescents attending the emergency department (ED). We aimed to describe patient characteristics and management of febrile adolescents attending the ED.

**Design and setting** The MOFICHE/PERFORM study (Management and Outcome of Febrile Children in Europe/Personalised Risk assessment in Febrile illness to Optimise Real-life Management across the European Union), a prospective multicentre study, took place at 12 European EDs. Descriptive and multivariable regression analyses were performed, comparing febrile adolescents (12–18 years) with younger children in terms of patient characteristics, markers of disease severity (vital signs, clinical alarming signs), management (diagnostic tests, therapy, admission) and diagnosis (focus, viral/bacterial infection).

**Results** 37 420 encounters were included, of which 2577 (6.9%) were adolescents. Adolescents were more often triaged as highly urgent (38.9% vs 34.5%) and described as ill appearing (23.1% vs 15.6%) than younger children. Increased work of breathing and a non-blanching rash were present less often in adolescents, while neurological signs were present more often (1% vs 0%). C reactive protein tests were performed more frequently in adolescents and were more often abnormal (adjusted OR (aOR) 1.7, 95% CI 1.5 to 1.9). Adolescents were more often diagnosed with SBI (OR 1.8, 95% CI 1.6 to 2.0) and sepsis/meningitis (OR 2.3, 95% CI 1.1 to 5.0) and were more frequently admitted (aOR 1.3, 95% CI 1.2 to 1.4) and treated with intravenous antibiotics (aOR 1.7, 95% CI 1.5 to 2.0).

**Conclusions** Although younger children presented to the ED more frequently, adolescents were more often diagnosed with SBI and sepsis/meningitis. Our data emphasise the importance of awareness of severe infections in adolescents.

## Strengths and limitations of this study

► Our study provides detailed data on a large number of adolescents attending the emergency department with fever.
► Our data show that, although accounting for a relatively small fraction of all emergency department visits for febrile children, adolescents have an increased risk of serious bacterial infections.
► Febrile adolescents are hospitalised more often, are more often treated with intravenous antibiotics and more often required immediate life-saving interventions.
► Adolescents with serious bacterial infections present differently from younger children and more research is needed to be able to provide detailed guidelines for this age group.

## INTRODUCTION

The adolescent age is often described as a health paradox because, on the one hand, it is a time of enhanced physical and mental capabilities, yet the overall mortality/morbidity rates increase significantly, often due to risk-taking behaviours such as substance abuse or injuries.[1] Although this health paradox has received considerable attention in the international literature, there is knowledge gap regarding infectious problems in adolescents, as most studies on adolescents have focused on topics typically associated with adolescence, such as violence or mental health,[2–4] while most studies on infections have focused on younger children.[5]

Case series as well as population studies have shown adolescents to have an increased risk of sepsis in comparison with younger

children,[6 7] and several studies showed adolescents with sepsis to have increased mortality rate.[6–11] One possible explanation for this increased case fatality rate might be the atypical presentation of adolescents with sepsis, such as gastrointestinal complaints.[9 12] The increased incidence of sepsis and the high case fatality rates emphasise the importance of awareness of severe infections in adolescents.

Despite this, little is known on the presentation, management and diagnosis of febrile adolescents presenting to the emergency department (ED). Our aim was to assess the presentation, management and diagnosis of febrile adolescents attending the ED and explore the differences between adolescents and younger children.

## METHODS

### Study design

This study is part of the MOFICHE study (Management and Outcome of Febrile Children in Europe), which is embedded in the PERFORM study (Personalised Risk assessment in Febrile illness to Optimise Real-life Management across the European Union).[13] MOFICHE is an observational multicentre study that evaluates the management and outcome of febrile children in Europe using routinely collected data.[14] In this substudy we specifically assessed patient characteristics, diagnosis and management of febrile adolescents and compared them with the characteristics and management of younger children.

### Patient and public involvement

Patients were not involved in the design of the study.

### Study population and setting

Twelve EDs from eight different countries (Austria, Germany, Greece, Latvia, the Netherlands (n=3), Spain, Slovenia and the UK (n=3)) participated in the study (online supplemental appendix 1). Participating hospitals were either tertiary university hospitals or large teaching hospitals (online supplemental appendix 2). Data were collected for at least 1 year (January 2017–April 2018).

For this analysis, the inclusion criteria were children aged 3 months to 18 years presenting to the ED with fever (temperature ≥38.0°C) or a history of fever in the previous 72 hours.

### Data collection

Data were obtained from patient records and entered into an electronic case report form. Data included general patient characteristics, such as age, sex, comorbidity, previous medical care, arrival time, referral (self, primary care physician, emergency medical services (EMS) or other), triage urgency, vital signs, presence of 'red traffic light' alarming signs from the National Institute for Health and Care Excellence (NICE) fever guideline[15] and high-risk criteria from the NICE sepsis guideline[15] (table 1), and management at the ED. The

**Table 1** Differences in patient characteristics between young children and adolescents (N=37 420)

| | Children 3 months–12 years n=34 843 n (%) | Children ≥12 years n=2577 n (%) |
|---|---|---|
| Male | 19 182 (55.1) | 1307 (50.7) |
| Age in years, median (IQR) | 2.6 (1.3–4.9) | 14.5 (13.2–16.1) |
| **Comorbidity†** | | |
| Simple | 4302 (12.5) | 489 (19.1) |
| Complex | 1332 (3.9) | 241 (9.4) |
| **Duration of fever*** | | |
| <24 hours | 11 410 (35.1) | 854 (37.3) |
| 24–48 hours | 10 622 (32.7) | 682 (29.8) |
| >48 hours | 10 433 (31.1) | 755 (33.0) |
| **Referral** | | |
| Self | 19 537 (57.8) | 1231 (49.4) |
| General practitioner/private paediatrician | 5654 (16.7) | 493 (19.8) |
| Emergency medical service | 5010 (14.8) | 430 (17.3) |
| Other | 3574 (10.6) | 337 (13.5) |
| **Triage urgency** | | |
| High: immediate, very urgent, intermediate | 11 664 (34.5) | 967 (38.9) |
| **Vital signs‡ and PEWS** | | |
| Tachycardia APLS | 8552 (24.5) | 764 (29.6) |
| Tachypnoea APLS | 5282 (15.2) | 189 (7) |
| Hypoxia, oxygen saturation <95% APLS | 805 (2.3) | 30 (1) |
| Prolonged capillary refill ≥3 s (ns) | 343 (1.1) | 25 (1) |
| Simplified PEWS 6 or higher | 782 (4.5) | 81 (6) |
| **NICE 'red traffic lights' (alarming signs)** | | |
| Ill appearance | 5203 (15.6) | 559 (23.1) |
| Increased work of breathing | 3050 (10.0) | 67 (3) |
| Rash: petechiae/non-blanching | 1040 (3.4) | 53 (2) |
| Decreased consciousness (ns) | 178 (1) | 16 (1) |
| Meningeal signs | 97 (0) | 23 (1) |
| Status epilepticus (ns) | 58 (0) | 8 (0) |
| Focal neurology | 110 (0) | 19 (1) |

Missing values: general patient characteristics: <7%; vital signs: 9%–23%; NICE alarming signs: 1%–18%.
All comparisons were p<0.001, unless otherwise indicated.
*P≤0.05.
†Comorbidity: a chronic underlying condition that is expected to last at least 1 year. Complex comorbidity: a chronic condition in ≥2 body systems or malignancy or immunocompromised patients.
‡According to APLS cut-off values by age.
APLS, Advanced Paediatric Life Support; NICE, National Institute for Health and Care Excellence; NS, not significant; PEWS, Paediatric Early Warning Scores.

NICE alarming signs include level of consciousness, ill appearance, increased work of breathing, age <3 months, non-blanching rash, meningeal signs, status epilepticus and focal neurological signs. The high-risk criteria from

**Table 2** 'Red traffic light' symptoms (alarming signs) from the NICE guideline on fever and high-risk criteria from the NICE sepsis guideline

| | Fever <5* | Sepsis <5† | Sepsis 5–11† | Sepsis >12† |
|---|---|---|---|---|
| **Behaviour** | | | | |
| No response to social cues‡ | + | + | | |
| Altered behaviour‡ | | | + | + |
| Ill appearance | + | + | + | |
| Does not wake/does not stay awake | + | + | + | |
| Weak, high-pitched or continuous cry§ | + | + | | |
| **Respiratory** | | | | |
| Grunting | + | + | | |
| Apnoea | | + | | |
| Oxygen saturation <90% | | + | + | |
| Oxygen saturation <93% | | | | + |
| Tachypnoea for age | + | + | + | + |
| Chest retractions | + | | | |
| **Circulation** | | | | |
| Bradycardia <60 | | + | + | |
| Tachycardia for age | | + | + | + |
| Reduced skin turgor | + | | | |
| Did not pass urine in the previous 18 hours | | | | + |
| Systolic blood pressure 90 mm Hg | | | | + |
| **Skin** | | | | |
| Mottled, ashen or cyanosis | + | + | + | + |
| Non-blanching rash | + | + | + | + |
| **Temperature** | | | | |
| <36.0°C | | + | | |
| ≥38.0°C in infants <3 months | + | + | | |
| **Neurological** | | | | |
| Bulging fontanelle or neck stiffness | + | | | |
| Status epilepticus | + | | | |
| Focal neurological signs | + | | | |
| Focal seizures | + | | | |

🟩-data available; 🟨-available (proxy used); 🟥-not available.
*NICE fever guideline.
†NICE sepsis guideline.
‡Defined as reduced consciousness.
§Defined as ill appearance.
NICE, National Institute for Health and Care Excellence.

the NICE sepsis guideline overlap with the NICE alarming signs but differ by age group, and include abnormal behaviour, decreased consciousness, low oxygen saturation, abnormal heart rate, respiratory rate or blood pressure, hypothermia, age <3 months, diminished urine output, cyanosis, or a non-blanching rash. An overview of the collected alarming signs is provided in table 2.

Data collection ranged from 1 week per month to the entire month, depending on the number of ED visits per hospital (online supplemental appendix 2).

Management comprised diagnostic tests (performance of blood tests, imaging, blood cultures and C reactive protein (CRP) test), treatment (intravenous antibiotics, oxygen, immediate life-saving interventions (ILSI)) and disposition (discharged, general ward admission or paediatric intensive care unit (PICU) admission).

### Definitions

Adolescents were defined as children aged 12–18 years; younger children were defined as children aged 3 months to 12 years.

Previous medical care was defined as medical care for the same complaint in the last 5 days at any facility, including a general practitioner. A previous ED visit was

defined as a visit to either the same or a different ED in the previous 5 days.

Comorbidity was defined as a chronic underlying condition that is expected to last at least 1 year.[16]

Vital signs were classified as abnormal according to Advanced Paediatric Life Support reference ranges.

Simplified Paediatric Early Warning Scores (PEWS) were calculated based on the PEWS developed by Parshuram *et al* (vital signs, capillary refill time, work of breathing and oxygen therapy, combined into a score).[17 18] Blood pressure was excluded from the PEWS as it was not routinely performed in our study. A previous study showed that a simplified PEWS without blood pressure showed similar performance in predicting PICU admission in comparison with the original PEWS.[18]

Triage categories were combined into 'low urgency' (non-urgent and standard) and 'high urgency' (urgent, very urgent and immediate).

ILSI[19] was categorised into the following: airway/breathing support, electrical therapy, emergency procedures, haemodynamic support and emergency medications (online supplemental appendix 3).

Focus of infection was categorised into upper respiratory, lower respiratory, gastrointestinal/surgical abdomen, urinary, skin, musculoskeletal, sepsis, meningitis/central nervous system (CNS), influenza-like illness, childhood exanthemas, inflammatory, undifferentiated fever or other.[14]

The consortium developed a consensus-based flow chart[14 20 21] to classify the presumed cause of infection for each visit (online supplemental appendix 4), depending on clinical signs and on CRP and microbiological tests (bacterial cultures, viral or bacterial PCR), into 'definite or probable bacterial', 'definite or probable viral', 'unknown', or 'other'.

Serious bacterial infection (SBI) was defined as 'definite/probable bacterial' with a focus on gastrointestinal, lower respiratory, urinary or musculoskeletal tract, CNS or sepsis. Sepsis/meningitis was defined as 'definite/probable bacterial' with a focus on CNS or sepsis.

### Data quality and missing data
Data quality and completeness were improved and standardised using a digital training module for physicians who assess febrile children at the ED, including clarification of the NICE alarming signs. Data were entered into the patient's record as part of routine care by the treating physician and nurse and were then manually extracted from these records and entered into an electronic case report form by trained research team members.

Missing determinants such as vital signs were handled by multiple imputation (table 1). Imputation was performed using the MICE package in R V.3.4. SPSS V.25 was used for data analysis.

### Data analysis
We performed descriptive analyses for general patient characteristics, vital signs, NICE alarming signs,

management, disposition and diagnosis. Characteristics of adolescents and younger children were compared using $\chi^2$ test and Mann-Whitney test. Results were deemed significant at $p<0.05$.

We analysed differences in management, disposition and presumed cause of infection by multivariable logistic regression, displayed as OR, and adjusted for general patient characteristics (setting/ED, sex, fever duration, previous medical care, arrival time and comorbidity), displayed as adjusted OR (aOR). We did not adjust for disease severity as our aim was to describe differences in disease severity between young children and adolescents. Subgroup analyses were performed for children and adolescents diagnosed with SBI and for children without comorbidity.

## RESULTS
### Patient characteristics
The study included 37 420 ED encounters, of which 2577 (6.9%) were adolescents (table 1). Adolescents were less often self-referred (49.4% vs 57.8%) and more often presented by EMS than younger children (17.3% vs 14.8%, p<0.001). In this study, 2816 (8.1%) younger children and 239 adolescents (9.3%) had attended an ED in the previous 5 days. Adolescents more often had comorbidity (28.5% vs 16.4%, p<0.001; table 1 and online supplemental appendix 5).

### Presenting signs and symptoms
Adolescents were more often triaged as highly urgent. Tachycardia was present more often (29.6% vs 24.5%, p<0.001), while tachypnoea, increased work of breathing and low oxygen saturation were present less often. Adolescents more often had PEWS of 6 or higher (6.2% vs 4.5%, p<0.001). Non-blanching rashes were present less often in adolescents, while ill appearance, meningeal signs and focal neurological signs were present more often (table 1).

In a subanalysis of children without comorbidity, the results were similar, except for triage urgency, PEWS and focal neurological signs, which were similar in both groups.

### Management
After adjusting for general patient characteristics, we found that diagnostic tests such as CRP were performed more often in adolescents (aOR 1.9, 95% CI 1.7 to 2.0) and that CRP more often reached levels >60 mg/L (aOR 1.7, 95% CI 1.5 to 1.9). Hospital admission (aOR 1.3, 95% CI 1.2 to 1.4), intravenous antibiotics (aOR 1.7, 95% CI 1.5 to 1.9) and ILSI (aOR 1.5, 95% 1.2 to 2.0) were more common in adolescents, while PICU admission was similar in both age groups (aOR 1.1, 95% CI 0.6 to 1.9; figures 1 and 2).

Of children who had attended the ED previously, 36.1% of younger children and 49.0% of adolescents were admitted (p<0.001). Intensive care unit admission

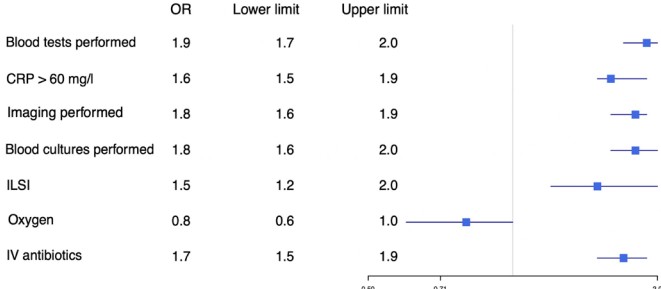

**Figure 1** Adjusted OR for diagnostic tests and therapy in younger children versus adolescents. Younger children were used as reference. Adjusted for hospital, sex, duration of fever, previous medical care, time of arrival and comorbidity. To convert CRP values to nmol/L, multiply by 0.9524. CRP, C reactive protein; ILSI, immediate life-saving intervention; IV, intravenous.

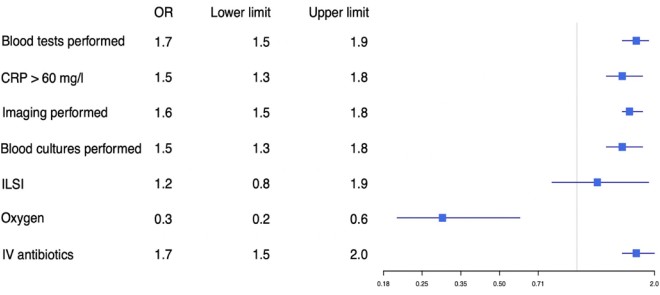

**Figure 3** Adjusted OR for diagnostic tests and therapy in younger children versus adolescents, patients with comorbidity excluded. Younger children were used as reference. Adjusted for hospital, sex, duration of fever, previous medical care and time of arrival. To convert CRP values to nmol/L, multiply by 0.9524. CRP, C reactive protein; ILSI, immediate life-saving intervention; IV, intravenous.

was similar for younger children (0.9%) and adolescents (0.8%). Subanalysis in children without comorbidity showed similar results, except for ILSI, which was similar in both groups (figures 3 and 4).

### Focus and presumed cause of infection

Upper respiratory tract infection was the most common focus in both age groups, although this was less common in adolescents than in younger children (41.8% vs 53.9%, p<0.001; figure 5). Gastrointestinal/surgical abdomen was diagnosed more often in adolescents (16.2% vs 10.1%, p<0.001). Adolescents were more often classified as having bacterial disease (31.0% vs 21.6%; aOR 1.5, 95% CI 1.4 to 1.7) and SBI (15.8% vs 8.4%; aOR 1.8, 95% CI 1.6 to 2.0) and less often with probable/definite viral disease. The most common SBIs in both adolescents and younger children were lower respiratory tract, urinary tract and gastrointestinal infections. Bacterial sepsis/meningitis was more common in adolescents (0.6% vs 0.3%; OR 1.9, 95% CI 1.1 to 3.3), although after adjusting for general patient characteristics this was significant only in children without comorbidity (aOR 2.3, 95% CI 1.1 to 5.0; figure 4).

Of children who had attended the ED previously, 13.8% of younger children and 21.8% of adolescents were diagnosed with SBI (p<0.001).

### Presentation and management of children and adolescents with SBI

In total, 3347 children presented with SBI. SBI was present in 406 of 2577 adolescents (15.8%) and 2941 of 34 843 younger children (8.4%).

Adolescents with SBI more often had comorbidity (34.0% vs 23.6%, p<0.001) and less often presented with tachypnoea and increased work of breathing, while rates of tachycardia and prolonged capillary refill were similar between adolescents and younger children with SBI. Adolescents with SBI and sepsis/meningitis were more often described as ill appearing, and adolescents who were described as ill appearing more often were diagnosed with SBI or sepsis/meningitis. However, the high-risk criteria from the NICE sepsis guideline were present less frequently in adolescents with SBI (online supplemental appendix 6).

No differences were found regarding the frequency of CRP >60 mg/L, intravenous antibiotics, admission or PICU admission. Adolescents with SBI were more often treated with ILSI than younger children (aOR 2.2, 95%

| | OR | Lower limit | Upper limit |
|---|---|---|---|
| Any admission | 1.3 | 1.2 | 1.4 |
| PICU admission | 1.1 | 0.6 | 1.9 |
| Admission with an intervention | 1.5 | 1.3 | 1.7 |
| SBI | 1.8 | 1,6 | 2.0 |
| Sepsis/meningitis | 1.5 | 0,9 | 2.6 |

**Figure 2** Adjusted OR for disposition and final diagnosis in younger children versus adolescents. Younger children were used as reference. Adjusted for hospital, sex, duration of fever, previous medical care, time of arrival and comorbidity. PICU, paediatric intensive care unit; SBI, serious bacterial infection.

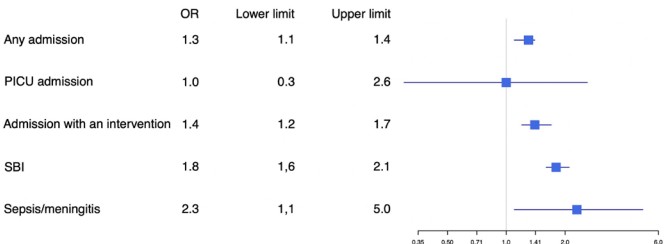

**Figure 4** Adjusted OR for disposition and final diagnosis in younger children versus adolescents, patients with comorbidity excluded. Younger children were used as reference. Adjusted for hospital, sex, duration of fever, previous medical care and time of arrival. PICU, paediatric intensive care unit; SBI, serious bacterial infection.

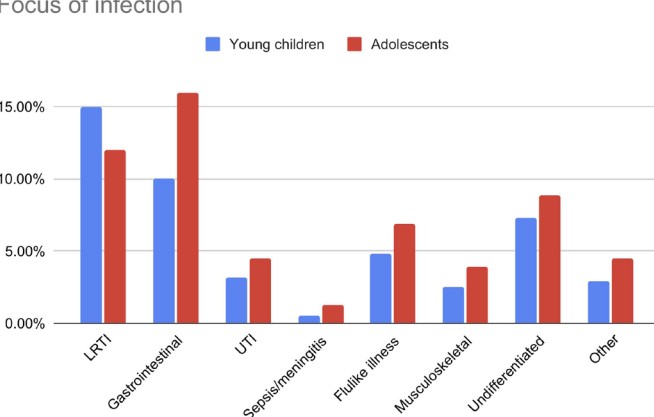

**Figure 5** Focus of infection in young children and adolescents. Data shown as percentages within the groups of young children and adolescents. Gastrointestinal: gastrointestinal and surgical abdomen; exanthemas: exanthemas and influenza-like illness; musculoskeletal: soft tissue, skin and musculoskeletal infection. LRTI (not shown in graphic): young children 54%, adolescents 42%. LRTI, lower respiratory tract infection; UTI, urinary tract infection.

CI 1.4 to 3.5), although this difference was not significant after excluding children with comorbidity.

## DISCUSSION
### Main findings
A well-known statement emphasises how 'children are not small adults';[22 23] however, our data show that adolescents are not big children either. Our data show that despite accounting for a small fraction of all ED visits for febrile children, adolescents presenting to the ED have an increased risk of SBIs, such as sepsis/meningitis. Furthermore, adolescents with SBI present differently from younger children. Although adolescents were more often described as ill appearing, the high-risk criteria from the NICE sepsis guideline were present less frequently in adolescents with SBI or sepsis/meningitis. Adolescents were hospitalised more often and more often received intravenous antibiotics and ILSI.

### Findings in relation to previous literature
Previous studies on febrile children have mainly focused on infants and young children[5] and literature regarding febrile adolescents is scarce. A recent study by Brockhus *et al*[23] on adolescents attending the ED showed that adolescents present with complaints different from those in children as well as adults and that infectious problems were far less common than trauma or mental health issues. However, as stated above, although adolescents present with infectious problems less common than younger children, adolescents that *do* present to the ED have an increased risk of suffering from a severe infection. Our data are in line with the few studies that have found an increased incidence of sepsis in adolescents in comparison with younger children[7 9] and emphasise how febrile adolescents form a distinct risk group. In line with

this, a study by Glynn and Moss[24] showed the severity of several infectious diseases (eg, varicella, mononucleosis, meningococcal infections and scarlet fever) to be high in infancy, lower in school-aged children and then again increasing from the adolescent age, following a J-shaped pattern.

To date it is unclear why adolescents have an increased risk of serious infections. Possible explanations include immunological deterioration with age, the so-called 'immune senescence', the influence of sex hormones on the immune system or the increase of comorbidities with age.[24] Regarding immune senescence, Glynn and Moss[24] suggest that this process starts earlier than previously believed, with optimal immune function being reached at age 5–14 and a decrease in immune function starting from adolescence. Evidence supporting this theory comes from data on vaccine response by age, showing a decreased vaccine response in adolescents.[24] Regarding puberty and hormonal influences, data seem to be inconsistent as the increased mortality rates in infectious diseases seen in males in comparison with females are not seen in adolescence.[24] Regarding comorbidity, although in our study adolescents more often had comorbidity, SBIs were still more common in adolescents after excluding children with comorbidity, and sepsis/meningitis was more common in adolescents only in the subgroup of children without comorbidity, showing that comorbidity does not offer a clear-cut explanation for these trends.

In addition to presenting with different rates of the same diseases, adolescents with the same disease can present differently as well.[12] As stated before, although not 'small adults', adolescents are not 'big children' either, differing from both adults and children with regard to physiology, immune system and endocrine system.[25] Further differences might be explained by differences in health-seeking behaviour, lack of parental supervision or a delayed presentation,[10 12 26] although the latter was not the case in our study. Regarding parental supervision, this was shown to be related to treatment adherence in adolescents with cystic fibrosis[27] or diabetes,[28] but there is a paucity of data regarding parental supervision in adolescents with infectious diseases and its impact on presentation or disease course.

Our data show that the *health paradox* does not only apply to preventable injuries, but also to potentially vaccine-preventable infections, such as meningococcal disease, or treatable infectious diseases, such as sepsis, where early recognition has the potential to improve outcome.[10 29]

### Implications for clinical practice and research
Our data highlight a gap in clinical guidelines addressing the presentation and management of febrile adolescents. While the NICE sepsis guideline addresses adolescents as well as younger children,[15] its focus is on the recognition of sepsis, while the more general NICE fever guideline is exclusively targeted at children below the age of 5, and our data show how the alarming signs for this age group cannot be extrapolated unambiguously to adolescents.[15]

Furthermore, as most studies on adolescents focused on mental health issues, there is a paucity of literature on febrile adolescents.

As adolescents with infectious problems form only a small fraction of ED visits,[23] exposure for each individual healthcare provider is expected to be low,[30] making the management of this group even more challenging. Therefore, there is an urgent need for future studies directed at identifying clinical criteria that can help improve the identification of SBI in the febrile adolescent age group.

While awaiting future studies, healthcare workers evaluating adolescents should have an increased level of awareness regarding the substantial risk of SBI in adolescents and the potentially different presentation. Diagnostic tests and antibiotic therapy should be considered at a low threshold when SBI cannot be ruled out on clinical grounds. In addition to ordering routine tests such as CRP and white cell count, clinicians should consider performing additional tests such as lactate and procalcitonin, as these offer improved diagnostic performance when it comes to differentiating sepsis from other causes of fever.[15 31–34]

Second, safety netting advice should be given to all febrile adolescents and their caregivers in case of discharge from the ED. Empowering adolescents on when to seek help and when and how to self-care at home is an important step in the management of febrile illnesses in adolescents. Most studies regarding patients' knowledge on fever have focused on caregivers of young children,[35 36] as do many online information sources.[37] Previous research on how to improve empowerment in adolescents with medical problems can aid in optimally addressing this specific population as this requires a different approach from addressing adults or parents of younger children.[38 39]

### Strengths and limitations

To our knowledge this is the first study looking into patient characteristics, management and diagnosis of febrile adolescents attending the ED.

The main strengths of our study are that detailed information on presenting signs, management and diagnosis was collected on a large number of children and adolescents in different European EDs. Data were collected year-round and included different hospitals with different patient case mixes, largely increasing the generalisability of the results.[14 40] Furthermore, we included a large number of children with SBI, as determined by a uniformly applied flow chart.

The main limitations include the lack of information regarding outcome after the ED visit, for example, 30-day morbidity and mortality, in the use of routinely collected data. To ensure data quality, all study sites were extensively trained on accurate documentation of patient characteristics and quality checks were performed regularly. The amount of missing data was limited and its effects were reduced by using multiple imputation.[41] Another limitation is that blood pressure, cyanosis and diminished urine output were not included in the data collection. A previous study showed that, although hypotension is associated with serious illness in children, its sensitivity is limited as routine measurement in *all* children attending the ED[42 43] and it is a late sign in children with sepsis in comparison with adults. On the other hand, in adolescents, similar to adults, hypotension might present earlier in the disease course and thus including blood pressure could provide valuable information.

As cases defined as 'probable bacterial' were also included in the definition of sepsis/meningitis, we cannot preclude that some of these cases were not of bacterial origin in either age group. However, in the European Union Childhood Life-threatening Infectious Disease Study (EUCLIDS) on severe sepsis, a pathogen was only found in half of the cases.[11] Lastly, our data apply to adolescents attending the ED; more research is needed to know whether our results can be applied to adolescents presenting to primary care as well. Furthermore, it is unknown whether a form of 'selection bias' exists as parents might be more inclined to seek help for younger febrile children than for adolescents.

### CONCLUSION

Our data show that despite accounting for a relatively small fraction of all ED visits, febrile adolescents have an increased risk of SBIs, including sepsis/meningitis, in comparison with younger children.

**Author affiliations**
[1]Department of Pediatrics, Erasmus MC Sophia Children's Hospital, Rotterdam, The Netherlands
[2]Institute of Infection and Global Health, University of Liverpool, Liverpool, UK
[3]Division of Paediatric Infectious Diseases, Munich University Hospital Dr von Hauner Children's Hospital, Munchen, Germany
[4]Clinical Research Department, London School of Hygiene and Tropical Medicine, London, UK
[5]Paediatric Immunology, Infectious Diseases and Allergy, Newcastle upon Tyne Hospitals NHS Foundation Trust, Great North Children's Hospital, Newcastle upon Tyne, UK
[6]Department of Paediatric Infectious Diseases and Immunology, Wilhelmina Children's Hospital, University Medical Centre Utrecht, Utrecht, The Netherlands
[7]Stichting Katholieke Universiteit, Radboudumc Nijmegen, Nijmegen, Netherlands
[8]Section of Paediatric Infectious Diseases, Imperial College London, London, UK
[9]Department of General Paediatrics, Medical University of Graz, Graz, Steiermark, Austria
[10]Genetics, Vaccines, Infections and Pediatrics Research group (GENVIP), Hospital Clínico Universitario de Santiago de Compostela, Santiago de Compostela, Spain
[11]Department of Infectious Diseases, University of Ljubljana, Ljubljana, Slovenia
[12]Department of Paediatric Infectious Diseases, National and Kapodistrian University of Athens, Athens, Greece
[13]Department of Paediatric Infectious Diseases and Immunology, Erasmus MC Sophia Children's Hospital, Rotterdam, The Netherlands
[14]Department of Pediatrics, Riga Stradins University, Riga, Latvia

**Acknowledgements** We gratefully acknowledge the emergency department staff of the participating hospitals for their participation and collection of data. In addition, we thank Arianne van Rijn, (former) medical student, for her assistance with literature review.

**Collaborators** Membership of the PERFORM Consortium is provided in online supplemental appendix 1.

**Contributors** Conceptualisation and design: DB, NNH, EC, UvB, JED, ME, MvdF, RdG, JH, BK, ML, EL, IM, FMT, RN, MP, IR-C, MT, DZ, WZ, HAM. Data acquisition: DB, NNH, EC, UvB, JED, ME, MvdF, RdG, JH, BK, ML, EL, IM, FMT, RN, MP, IR-C, MT, CV, DZ, WZ, JZ, HAM. Data verification: DB, NNH. Data interpretation: DB, NNH, HAM. Formal analysis: DB, NNH. Writing - original manuscript: DB. Critically revising the manuscript: DB, NNH, EC, UvB, JED, ME, MvdF, RdG, JH, BK, ML, EL, IM, FMT, RN, MP, IR-C, MT, CV, DZ, WZ, JZ, HAM. The corresponding author attests that all listed authors meet the authorship criteria and that no others meeting the criteria have been omitted. This publication is the work of the authors, who will serve as guarantors for the content of this paper.

**Funding** This project has received funding from the European Union's Horizon 2020 research and innovation programme under grant agreement number 668303. The research was supported by the National Institute for Health Research Biomedical Research Centre based at Imperial College (JH, ML) and at Newcastle Hospitals NHS Foundation Trust and Newcastle University (EL, ME).

**Competing interests** None declared.

**Patient consent for publication** Not required.

**Ethics approval** The study was approved by the ethical committees of all the participating hospitals and no informed consent was needed for this study. Austria: Ethikkommission Medizinische Universitat Graz (ID: 28-518ex15/16); Germany: Ethikkommission Bei Der LMU München (ID: 699-16); Greece: Ethics Committee (ID: 9683/18.07.2016); Latvia: Centrala medicinas etikas komiteja (ID: 14.07.201.6.No. Il16-07-14); Slovenia: Republic of Slovenia National Medical Ethics Committee (ID: 0120-483/2016-3); Spain: Comité Autonómico de Ética de la Investigación de Galicia (ID: 2016/331); the Netherlands: Commissie Mensgebonden onderzoek (ID: NL58103.091.16); UK: Ethics Committee (ID: 16/LO/1684, IRAS application number 209035; confidentiality advisory group reference: 16/CAG/0136). In all the participating UK settings, an additional opt-out mechanism was in place.

**Provenance and peer review** Not commissioned; externally peer reviewed.

**Data availability statement** Data are available upon reasonable request. Individual participant data that underlie the results reported in this article, including a data dictionary, will be made available after de-identification to researchers who provide a methodologically sound proposal. Proposals should be directed to d.borensztajn@erasmusmc.nl. To gain access, data requestors will need to sign a data access agreement.

**ORCID iDs**
Dorine Borensztajn http://orcid.org/0000-0002-2437-0757
Nienke N Hagedoorn http://orcid.org/0000-0001-9237-4904
Ulrich von Both http://orcid.org/0000-0001-8411-1071
Juan Emmanuel Dewez http://orcid.org/0000-0002-5677-8968
Henriette A Moll http://orcid.org/0000-0001-9304-3322

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
