## [Reviewer comments · BMJ Open]

ARTICLE DETAILS

TITLE (PROVISIONAL)	The adolescent paradox at the ED: teenagers visiting the ED have an increased risk of serious bacterial infections.
AUTHORS	Borensztajn, Dorine; Hagedoorn, Nienke; Carrol, Enitan; von Both, Ulrich; Dewez, Juan; Emonts, Marieke; van der Flier, Michiel; de Groot, Ronald; Herberg, Jethro; Kohlmaier, Benno; Levin, M; Lim, Emma; Maconochie, Ian; Martinon Torres, Federico; Nijman, R; Pokorn, Marko; Rivero-Calle, Irene; Tsolia, Maria; Vermont, Clementien; Zavadska, Dace; Zenz, Werner; Zachariasse, Joany; moll, Henriette

VERSION 1 – REVIEW

REVIEWER	Pek, Jen Heng Sengkang General Hospital, Acute and emergency care centre
REVIEW RETURNED	08-Jul-2021

GENERAL COMMENTS	I thank the authors for this opportunity to review their work. This is interesting and relevant to the ED setting especially since sepsis can be difficult to diagnose in the paediatric population. I think what is currently missing in the manuscript is an more indepth discussion of the study's results. For instance,  1) Why do adolescents have an increased risk of SBI? 2) Why do they present differently? 3) What should emergency department do to identify the patients with sepsis/SBI since guidelines like high risk criteria from NICE sepsis perform poorly? 4) What challenges exist for the management of febrile illness in the adolescents? The authors should be able to make some postulations and recommendations based on their data collected. My other regret is that the authors were unable to include the outcome of these febrile episodes in the study population, making the picture a less than complete one.
--

REVIEWER	Stover, Kayla University of Mississippi School of Pharmacy, Pharmacy Practice
REVIEW RETURNED	09-Aug-2021

GENERAL COMMENTS	This manuscript describes a portion of results from the PERFORM trial, focused on adolescents vs. younger children presenting to emergency departments across the European Union. This manuscript provides additional information to the literature to supplement the numerous studies related to psychological health or concerns. A few considerations are listed below:
--

	 1. The formatting throughout could be improved to enhance readability (paragraphs instead of individual statements). 2. Define all abbreviations the first time used. 3. Study objectives could be more clearly stated in the methods. 4. Discussion: (p. 18, lines 25-28): authors could spend additional time discussing why the health paradox applies to "potentially preventable infectious diseases". What makes these diseases described here "potentially preventable"? Why do you think adolescents are particularly at risk vs. younger children? 5. Discussion (p. 19, lines 10-16): since this study focuses on adolescents, authors should expand this discussion to empowering/educating adolescents on when to seek care/home options for treatment/managing early symptoms, etc. (A portion of this population may be self-responsible for care). 6. Figures 1 and 3 are difficult to read as presented. Perhaps the font could be increased or formatting altered to improve readability.
--	---

VERSION 1 – AUTHOR RESPONSE

Reviewer: 1

Dr. Jen Heng Pek, Sengkang General Hospital

Comments to the Author:

I thank the authors for this opportunity to review their work. This is interesting and relevant to the ED setting especially since sepsis can be difficult to diagnose in the paediatric population.

Thank you very much for taking the time to review our manuscript and providing us with very useful suggestions to improve our manuscript.

I think what is currently missing in the manuscript is a more in-depth discussion of the study's results. For instance,

1) Why do adolescents have an increased risk of SBI?

We thank the reviewer for this question and have added the following paragraphs to the discussion:

“In line with this, a study by Glynn et al [Glynn 2020] showed the severity of several infectious diseases (e.g. varicella, mononucleosis, meningococcal meningitis and scarlet fever) to be high in infancy, lower in school-aged children and then again increasing from the adolescent age, following a J-shaped pattern.

To date it is unclear why adolescents have an increased risk for serious infections. Possible explanations include immunological deterioration with age, so-called “immune senescence”, the influence of sex hormones on the immune system or the increase of comorbidity with age. [Glynn 2020] Regarding immune senescence, Glynn et al suggest that this process starts earlier than previously believed, with optimal immune function being reached at age 5-14, and a decrease in immune function starting from adolescence. [Glynn 2020] Evidence supporting this theory comes from data on vaccine response by age, showing a decreased vaccine response in adolescents. [Glynn 2020] Regarding puberty and hormonal influences, data seems to be inconsistent as the increased mortality rates in infectious diseases seen in males in comparison to females is not seen in adolescence. [Glynn 2020] Regarding comorbidity, although in our study adolescents more often had comorbidity, SBI were still more common in adolescents after excluding children with comorbidity and

sepsis/meningitis was more common in adolescents only in the subgroup of children without comorbidity, showing that comorbidity does not offer a clear-cut explanation for these trends.”

2) Why do they present differently?

We thank the reviewer for this comment and have explored this matter by adding the following paragraph to the discussion.

“In addition to presenting with different rates of the same diseases, adolescents with the same disease can present differently as well. [Thompson 2006] As stated before, although not “small adults”, adolescents are not “large children” either, differing from both adults and children with regards to physiology, the immune system and the endocrine system. [WHO] Further differences might be explained by differences in health seeking behavior, lack of parental supervision or a delayed presentation [Thompson 2006] [Burman 2019] [Trivedi 2019] although the latter was not the case in our study. Regarding parental supervision, this was shown to be related to treatment adherence in adolescents with cystic fibrosis [Modi 2008] or diabetes, [Ellis 2007] but there is a paucity of data regarding parental supervision in adolescents with infectious diseases and its impact on presentation or disease course.”

3) What should emergency department do to identify the patients with sepsis/SBI since guidelines like high-risk criteria from NICE sepsis perform poorly?

4) What challenges exist for the management of febrile illness in the adolescents? The authors should be able to make some postulations and recommendations based on their data collected.

We thank the reviewer for highlighting these questions regarding challenges and clinical implications.

We have adjusted the discussion and it now reads as follows:

“Our data highlight a gap in clinical guidelines addressing the presentation and management of febrile adolescents. While the NICE sepsis guideline addresses adolescents as well as younger children, [Nice guideline] its focus is on the recognition of sepsis, while the more general NICE fever guideline is exclusively targeted at children below the age of five and our data show how the alarming signs for this age group cannot be extrapolated unambiguously to adolescents. [Nice guideline] Furthermore, as most studies on adolescents focus on mental health issues, there’s a paucity of literature on febrile adolescents.

As adolescents with infectious problems form only a small fraction of ED visits [Brockhus 2020] exposure for each individual health care provider is expected to be low [Evans 2017], making the management of this group even more challenging. Therefore, there’s an urgent need for future studies directed at identifying clinical criteria that can help improve the identification of SBI in the adolescent age group.

While awaiting future studies, health care workers evaluating adolescents should have an increased level of awareness regarding the substantial risk in adolescents for SBI and the potentially different presentation. Diagnostic tests and antibiotic therapy should be considered at a low threshold when SBI cannot be ruled out on clinical grounds. In addition to ordering routine tests such as C-reactive protein and white blood count, clinicians should consider performing additional tests such as lactate and procalcitonin, as these offer improved diagnostic performance when it comes to differentiating sepsis from other causes of fever. [Bell 2015] [Trippella 2017] [Hubert-Dibon 2018] [Karon 2017] [Nice guideline]

Secondly, safety netting advice should be given to all febrile adolescents and their caregivers in case of discharge from the ED. Empowering adolescents on when to seek help and when and how to self-care at home is an important step in the management of febrile illnesses in adolescents. Most studies

regarding patient's knowledge on fever have focused on caregivers of young children, [van de Maat 2018] [Thompson 2020] as do many online information sources. [NHS fever] Previous research on how to improve empowerment in adolescents with medical problems can aid in optimally addressing this specific population as this requires a different approach than addressing adults or parents of younger children. [Sinha 2021] [Albritton 2003]"

5. My other regret is that the authors were unable to include the outcome of these febrile episodes in the study population, making the picture a less than complete one.

We agree with the author that this is a limitation of our study and have listed this in the "Strengths and limitations" section.

"The main limitations include the lack of information regarding outcome after the ED visit, e.g. 30-day morbidity and mortality in the use of routinely collected data."

Furthermore, we have added an additional analysis regarding revisits.

We have added the following sentence to the methods section:

"A previous ED visit was defined as a visit to either the same or a different ED in the previous five days. "

We have added the following paragraph to the results section:

"2,816 (8.1%) younger children and 239 adolescents (9.3%) had attended an ED in the previous five days. Of those children that had attended the ED previously, 36.1% of younger children and 49.0% of adolescents were admitted ($p < 0.001$). ICU admission was similar for younger children (0.9%) and adolescents (0.8%). Of those children that had attended the ED previously, 13.8% of younger children and 21.8% of adolescents were diagnosed with SBI ($p < 0.001$)."

Reviewer: 1

Competing interests of Reviewer: None

Reviewer: 2

Dr. Kayla Stover, University of Mississippi School of Pharmacy

Comments to the Author:

This manuscript describes a portion of results from the PERFORM trial, focused on adolescents vs. younger children presenting to emergency departments across the European Union. This manuscript provides additional information to the literature to supplement the numerous studies related to psychological health or concerns. A few considerations are listed below:

1. The formatting throughout could be improved to enhance readability (paragraphs instead of individual statements).

Thank you for pointing this out, we have adjusted this throughout the manuscript.

2. Define all abbreviations the first time used.

We apologize for this omission and have corrected this matter.

3. Study objectives could be more clearly stated in the methods.

We thank the reviewer for addressing this point and have added the following sentences to the methods section.

“In this sub-study we specifically assessed the presentation, management and diagnosis of febrile adolescents and compared it to the characteristics and management of younger children.”

“Management comprised diagnostic tests (performance of blood tests, imaging, blood cultures and CRP test results), treatment (intravenous antibiotics, oxygen, Immediate Life Saving Interventions (ILSI)) and disposition (discharge, general ward admission or Paediatric Intensive Care unit (PICU) admission).”

4. Discussion: (p. 18, lines 25-28): authors could spend additional time discussing why the health paradox applies to "potentially preventable infectious diseases". What makes these diseases described here "potentially preventable"?

Thank you for pointing out this point was not clear enough. We have changed this statement into

“Our data show that the health paradox does not only apply to preventable injuries, but also to potentially vaccine-preventable infections, such as meningococcal disease, or treatable infectious diseases, such as sepsis, where early recognition has the potential to improve outcome. [Burman 2019][Hilarius 2020]”

5. Why do you think adolescents are particularly at risk vs. younger children?

We thank the reviewer for this question and have added the following paragraphs to the discussion: See also comment 1 made by reviewer number 1.

“In line with this, a study by Glynn et al [Glynn 2020] showed the severity of several infectious diseases (e.g. varicella, mononucleosis, meningococcal meningitis and scarlet fever) to be high in infancy, lower in school-aged children and then again increasing from the adolescent age, following a J-shaped pattern.

To date it is unclear why adolescents have an increased risk for serious infections. Possible explanations include immunological deterioration with age, so-called “immune senescence”, the influence of sex hormones on the immune system or the increase of comorbidity with age. [Glynn 2020] Regarding immune senescence, Glynn et al suggest that this process starts earlier than previously believed, with optimal immune function being reached at age 5-14, and a decrease in immune function starting from adolescence. [Glynn 2020] Evidence supporting this theory comes from data on vaccine response by age, showing a decreased vaccine response in adolescents. [Glynn 2020] Regarding puberty and hormonal influences, data seems to be inconsistent as the increased mortality rates in infectious diseases seen in males in comparison to females is not seen in adolescence. [Glynn 2020] Regarding comorbidity, although in our study adolescents more often had comorbidity, SBI were still more common in adolescents after excluding children with comorbidity and sepsis/meningitis was more common in adolescents only in the subgroup of children without comorbidity, showing that comorbidity does not offer a clear-cut explanation for these trends. “

5. Discussion (p. 19, lines 10-16): since this study focuses on adolescents, authors should expand this discussion to empowering/educating adolescents on when to seek care/home options for treatment/managing early symptoms, etc. (A portion of this population may be self-responsible for care).

We thank the reviewer for raising this important topic and have added the following lines to the discussion

See also comments 3 and 4 made by reviewer number 1.

“Secondly, safety netting advice should be given to all febrile adolescents and their caregivers in case of discharge from the ED. Empowering adolescents on when to seek help and when and how to self-care at home is an important step in the management of febrile illnesses in adolescents. Most studies regarding patient’s knowledge on fever have focused on caregivers of young children, [van de Maat 2018] [Thompson 2020] as do many online information sources. [NHS fever] Previous research on how to improve empowerment in adolescents with medical problems can aid in optimally addressing this specific population as this requires a different approach than addressing adults or parents of younger children. [Sinha 2021] [Albritton 2003]”

6. Figures 1 and 3 are difficult to read as presented. Perhaps the font could be increased or formatting altered to improve readability.

We have addressed this issue and hope you will find the updated images have improved.

Reviewer: 2

Competing interests of Reviewer: None

VERSION 2 – REVIEW

REVIEWER	Stover, Kayla University of Mississippi School of Pharmacy, Pharmacy Practice
REVIEW RETURNED	10-Dec-2021
GENERAL COMMENTS	Thank you for addressing these points.